# Molecular Mechanisms of Heat Shock Factors in Cancer

**DOI:** 10.3390/cells9051202

**Published:** 2020-05-12

**Authors:** Mikael Christer Puustinen, Lea Sistonen

**Affiliations:** 1Cell Biology, Faculty of Science and Engineering, Åbo Akademi University, Tykistökatu 6, 20520 Turku, Finland; mikael.puustinen@abo.fi; 2Turku Bioscience, University of Turku and Åbo Akademi University, Tykistökatu 6, 20520 Turku, Finland

**Keywords:** carcinogenesis, heat shock factor, heat shock response, proteotoxic stress, transcription

## Abstract

Malignant transformation is accompanied by alterations in the key cellular pathways that regulate development, metabolism, proliferation and motility as well as stress resilience. The members of the transcription factor family, called heat shock factors (HSFs), have been shown to play important roles in all of these biological processes, and in the past decade it has become evident that their activities are rewired during tumorigenesis. This review focuses on the expression patterns and functions of HSF1, HSF2, and HSF4 in specific cancer types, highlighting the mechanisms by which the regulatory functions of these transcription factors are modulated. Recently developed therapeutic approaches that target HSFs are also discussed.

## 1. Introduction

Heat shock factors, HSFs, both individually and cooperatively, regulate transcriptional programs that enable cellular adaptation to proteotoxic stress conditions and physiological changes during developmental and differentiation-related processes, especially those driving neurogenesis and gametogenesis [1,2]. Interestingly, functions of HSFs can be traced back to the original findings by Italian geneticist Ferruccio Ritossa, who found that exposures to elevated temperatures induce specific chromosome puffs in the salivary glands of *Drosophila busckii* larvae [3]. Further investigations showed that these chromosome puffs designated the specific genomic regions that code for heat shock proteins (HSPs), which function as molecular chaperones [4]. Subsequently, HSF1 was identified as a major transcription factor regulating the stress-inducible expression of HSPs, thereby driving this evolutionary well-conserved response, termed the heat shock response.

Since the initial discovery of HSF1 in invertebrates, additional HSFs have been identified in vertebrates [5,6,7]. During the past decade, high resolution genome-wide sequencing techniques have enabled mapping of the genomic loci to which HSFs bind and regulate the transcription of their target genes [8,9]. Based on these analyses, it has become evident that HSFs, individually and in cooperation with each other, regulate distinct transcriptional programs under various biological conditions, including developmental processes and pathologies [10,11]. Some of the disease-specific programs are linked to malignances such as cancer, where the altered levels of HSPs and other targets contribute to disease progression. In this review, we focus on the expression patterns and functions of HSF1, HSF2, and HSF4 in specific cancer types and on the key mechanisms regulating HSF-driven tumor progression. Moreover, we highlight the therapeutic approaches that have been recently developed to target HSFs in cancer.

## 2. Domain Structure in HSFs

Unlike a single HSF in invertebrates, six HSFs have been found in mammals (HSF1, HSF2, HSF3, HSF4, HSFX, and HSFY), and plants are known to express more than thirty different HSFs (for comprehensive reviews see [2,12,13]). Among the five human HSFs, the functions of HSF1, HSF2, and HSF4 have been most extensively studied [1,14,15]. Since the structural domains in HSF proteins have been described in great detail in a recent review [2], they are only briefly summarized here. All HSFs share a well-conserved modular structure and are classified based on the high amino acid sequence homology in the *N*-terminal DNA-binding domain (DBD) (Figure 1). The DBD allows them to bind to the consensus cis-acting heat shock elements (HSEs) in the target promoters. An HSE consists of inverted pentameric nGAAn repeats, where “n” can be any nucleotide, and usually more than two HSE motifs are required for HSF binding, whereas the spacing and orientation can vary [16]. However, it is important to note that the chromatin landscape has a fundamental impact on HSF binding. Another conserved domain among HSFs is the oligomerization domain (HR-A/B), consisting of leucine zipper-like heptad repeats, and this domain is functionally essential for the formation of HSF trimers [17]. Trimerization is a key step in the activation of HSFs, and heterotrimers of HSF1 and HSF2 have a different trans-activating capacity than homotrimers [18,19]. Spontaneous trimerization of HSFs under non-stress conditions is inhibited by a repressor motif called the HR-C domain [20]. Human HSF4 lacks an HR-C domain, which results in the formation of constantly active trimers [7]. The ability of HSFs to induce transcription is dependent on the activation domain (AD), which is rich in hydrophobic and acidic amino acid residues that make up the contact sites with the basal transcription complex [21,22]. HSF1 also contains a regulatory domain (RD), which represses the AD under normal conditions and facilitates activation upon cellular stress. To date, the functional relevance of the RD in HSF1 has been investigated in mammalian cells (for a comprehensive review, see [15]), whereas the corresponding regions in HSF2 and HSF4 are poorly characterized.

## 3. Activation and Attenuation

Multiple mechanisms have been reported to increase or decrease the trans-activating capacity of HSFs under non-stress and stress conditions, and collectively they form an HSF activation–attenuation cycle [13].

### 3.1. Molecular Mechanisms Regulating HSF Activation

A variety of proteotoxic stimuli, such as elevated temperatures, toxins and pathogens, activate an inert pool of HSF molecules in order to induce the transcription of stress-responsive genes, which initiates the heat shock response. Although the heat shock response appears to be an important defense response in most organisms, the primary mechanism initiating the response still remains unknown. Nonetheless, constitutive nuclear import coupled with stress-regulated nucleocytoplasmic shuttling plays a key role in modulating the amounts of HSF1 in the nucleus [23]. Therefore, it is likely that multiple intrinsic and extrinsic signals contribute to HSF activation. Several studies have shown that purified HSF proteins have an intrinsic capacity to form active complexes in vitro under different protein-damaging conditions [24,25,26]. In HSF1, the RD is essential for stress-responsive activation [27]. Furthermore, considering that the RD harbors 15 known phosphorylation sites, it has been suggested that stress-inducible phosphorylation within this domain facilitates HSF1 activation [28]. This mode of activation is, however, contradicted by findings showing that removal of these 15 phosphorylation sites within the RD neither results in spontaneous activation of HSF1 nor reduces its trans-activating capacity [29].

Currently, the most prominent model of HSF1 activation is termed the chaperone titration model. According to this model, HSF1 is sequestered and inactivated by molecular chaperones such as HSP40/*DNAJB1*, HSP70/*HSPA1A*, and HSP90/*HSPC* [30]. In cells exposed to proteotoxic stress, these chaperones are recruited to denatured proteins and as a result HSF1 is released from the chaperone complexes that keep it inert. The chaperone titration model is further supported by results showing that pharmaceutical inhibition of HSP90/*HSPC* in mammalian cells activates the heat shock response [31]. In the nematode *Caenorhabditis elegans*, the loss of Hsp-70 consequently results in robust HSF-1 activation [32], whereas in *Saccharomyces cerevisiae* HSF binds to the cytosolic HSP70 family members Ssa1 and Ssa2 under non-stress conditions and this interaction is disrupted by heat shock [33,34]. Zeng and colleagues also found that overexpressing Ssa2, together with its co-chaperone, attenuated yeast HSF activity [33]. Moreover, HSP70/*HSPA1A* and HSP40/*DNAJB1* can attenuate HSF1 activity when overexpressed in mammalian cells [35]. Intriguingly, in *C. elegans* HSF-1 activation in distal tissues can be modulated through specific thermosensory neurons [36]. Apart from heat shock, it has been shown that non-stress activation of the thermosensory neurons, by optogenetic stimulation, induces serotonin release which in turn activates the heat shock response in distal tissues [37]. To date, this type of activation mechanism has been identified only in nematodes.

Under normal growth conditions, the majority of HSF1 molecules reside as inert monomers in the cytoplasm. Nevertheless, some HSF1 trimers are present in the nucleus, where they bind to their target gene promoters and maintain basal gene expression [25,38,39]. The core promoter region of HSP genes is pre-loaded with the paused RNA polymerase II (Pol II), and the interaction between HSF1, replication protein A (RPA) and histone chaperone FACT keeps the promotor region open [9,40]. In response to stress, HSF1 trimers accumulate rapidly in the nucleus, where they induce the transcription of target genes [41,42]. Promoter-bound HSF1 trimers interact with general transcription factors and co-activators, including chromatin-remodeling complexes and histone-modifying enzymes, which open up the structure of chromatin, thereby facilitating the release of Pol II from the paused state into elongation (for a comprehensive review, see [43] and references therein). The multi-subunit co-activator mediator complex facilitates HSF1-mediated transcription by directly interacting with components of the transcription machinery and promoting assembly of the pre-initiation complex [22,44,45].

### 3.2. Molecular Mechanisms Regulating HSF Attenuation

Detailed mechanisms responsible for attenuating the heat shock response remain unknown. However, the chaperone titration model has been suggested to also modulate the attenuation phase. Accordingly, the stress-induced accumulation of HSPs facilitates their binding to HSF1, which forces de-activation of HSF1 trans-activating capacity. In addition, increased amount of HSP70/*HSPA1A* stimulates enrichment of the transcriptional co-repressor CoREST to the HSP70/*HSPA1A* promoter, which suppresses HSF1 activity [46]. Furthermore, a recent study revealed that HSP90/*HSPC* is involved in the clearing of HSF1 from the DNA as HSP90/*HSPC* inhibition both increased the duration of HSF1 binding to the HSP70/*HSPA1A* promoter and enhanced HSF1 trans-activating capacity in heat-shocked cells [47]. Another mechanism contributing to the attenuation of HSF1 is acetylation of the DBD, which was shown to stimulate the dissociation of HSF1 from DNA [48]. Acetylation in the DBD relies in part on activating transcription factor 1 (ATF1), which recruits the acetyl transferase complex CBP/p300 to the DNA-bound HSF1 [38]. However, the histone de-acetylases SIRT1, HDAC7, and HDAC9 can reduce acetylation of HSF1, thereby prolonging the activation phase of the heat shock response [48,49]. The attenuation phase is affected by the ubiquitin-mediated proteasomal degradation of HSF1, which is mediated by FBXW7, a subunit of the SCF (Skp1-Cul1-F box) ubiquitin E3 ligase complex [50]. In MCF-7 breast cancer cells, the serine/threonine-protein kinase PIM2 phosphorylates HSF1 at threonine 120, which disrupts the interaction between FBXW7 and HSF1 and increases HSF1 protein stability [51]. In certain cancer types, such as multiple myeloma, FBXW7 is downregulated, which causes accumulation of HSF1 to the nucleus and subsequently prevents the attenuation of the heat shock response [50]. In addition to HSF1, HSF2 activity is attenuated through proteasomal degradation, as HSF2 has been shown to be ubiquitinated by the ubiquitin E3 ligase anaphase-promoting complex/cyclosome (APC/C) in heat-shocked cells [52].

## 4. HSFs as Developmental Factors

HSFs are key molecules in both stressed and unstressed conditions and they have been demonstrated to be indispensable for cell viability in yeasts [53]. Interestingly, the original finding, linking HSFs to development, demonstrated that HSF1 is required for early larval development in the fruit fly *D. melanogaster* [54]. A more recent groundbreaking study revealed that HSF-1 is an essential developmental factor in *C. elegans* [11]. This analysis of the genome-wide occupancy of HSF-1 during larval development provided evidence for a specific set of HSF-1 target genes critical for nematode development that is different from those occupied by HSF-1 in response to acute heat stress [11]. However, some target genes overlap in the HSF-1-regulated gene networks under physiological and stress conditions. During larval development, the co-activator EFL-1/DPL-1 supports HSF-1 activity at the developmental target genes, demonstrating the importance of cooperation between factors regulating distinct transcriptional programs. Studies using HSF-null mice have made it possible to further identify the physiological and developmental processes in which HSF1, HSF2, and HSF4 are involved. All HSF-null mice are viable but display distinct phenotypes, such as abnormalities in sensory organs and the brain, while others have similar phenotypes, such as impaired fertility, for details, see below [14,55,56]. Of note, HSF-null mice are regarded as a representative model, since murine HSFs display more than 84% identical amino acid sequence to human HSFs. Importantly, genomic mapping shows that the segments of the chromosomes encoding HSFs are homologues between mice and humans [57].

### 4.1. HSFs in Fertility

Both HSF1 and HSF2 have been shown to be involved in regulating gametogenesis. HSF2-null mice display grave defects in gametogenesis, since males have increased apoptosis of spermatocytes and females have abnormal production of egg cells [58]. In testis, HSF2 was shown to regulate the expression of HSPs and Y chromosomal multi-copy genes, including *SSTY2*, *SLY*, and *SLX*, which have critical functions in determining sperm quality [59]. Consequently, when a proper HSF2 expression pattern was disrupted in the distinct phases of spermatogenesis, the male germ cells were unable to undergo maturation in HSF2-null mice [60]. Analysis of female HSF1-null mice indicated that although these mice produced oocytes they did not develop properly beyond the zygotic stage [61]. Furthermore, the lack of HSF1 was reported to contribute to abnormal development of the chorioallantoic placenta, which in turn increased prenatal lethality [62]. Interestingly, the double knockout of HSF1 and HSF2 resulted in male sterility, implying that the loss of both HSFs has an additive effect on spermatogenesis [63]. This finding is supported by genome-wide analyses in spermatogenic cells, showing that at physiological temperature, both HSF1 and HSF2 occupy the same genomic regions [64].

### 4.2. HSFs in Brain Development and Sensory Organs

HSF2 has important functions in brain development, since HSF2-null mice display brain abnormalities as characterized by enlarged ventricles, small hippocampus, and mispositioning of neurons [56,65]. Mechanistically, HSF2 modulates the neuronal migration by regulating the transcription of p35 and p39, upstream activators of cyclin-dependent kinase 5 (Cdk5), which is required for cortical lamination [65,66,67]. The importance of HSF2 in neuronal migration is further supported by studies on cellular and mouse models for fetal alcohol spectrum disorder [68]. In response to alcohol stress, HSF1 and HSF2 predominantly form heterotrimers that drive the stress-inducible expression of genes enhancing survival at the expense of genes involved in neuronal migration.

HSF4 has a unique function in lens development [14,69]. Early-onset lens deterioration is a common phenotype in HSF4-null mice, due to abnormal proliferation and differentiation of lens epithelial cells, which eventually results in post-natal cataract and blindness [69]. A central function of HSF4 is to promote the transcription of genes encoding crystallins, which are major structural proteins in the lens [70,71,72]. In primary human lens epithelial cells, HSF4 has been shown to directly bind to and stabilize p53, subsequently resulting in cell cycle arrest and reduced proliferation [73]. Furthermore, the importance of HSF4 in human lens development is highlighted by reports showing that a number of mutations in the DNA-binding domain of HSF4 are related to severe cases of chromosomal dominant hereditary cataract [74,75,76]. HSF4 is also involved in olfactory neurogenesis by modulating the expression of leukemia inhibitory factor (LIF), a key cytokine involved in normal development of olfactory sensory neurons [56,77]. In contrast to HSF4, HSF1 suppresses LIF expression, which likely explains why HSF1-null mice also display abnormalities in the olfactory epithelium [56].

## 5. HSFs in Cancer

Extensive long-term studies have suggested HSPs as important pro-survival and antiapoptotic proteins facilitating malignant transformation. Many of these studies have prompted the development of pharmacological inhibitors targeting HSPs, but no chemical inhibitors have yet received FDA approval [78]. Currently, there is a strong focus on HSFs as potential therapeutic targets, partly due to their regulatory role in transcription of HSP-encoding genes but also due to a multitude of newly discovered cancer-specific HSF target genes. In 2007, two groundbreaking studies demonstrated the importance of HSF1 in cancer [79,80]. Dai and colleagues found that HSF1-null mice displayed significant resistance to dimethylbenzanthracene (DMBA)-induced skin carcinogenesis and mutant p53-induced tumorigenesis [79]. Subsequently, HSF1-null mice were also shown to be resistant to carcinogen-induced liver cancer and HER2/ErbB2-induced breast cancer [81,82]. The second study published in 2007 reported that HSF1 abolishment in p53-null mice reduced the development of lymphomas but the formation of carcinomas and sarcomas was enhanced [80]. A similar shift in tumor development has been observed in p53 and HSF4 double-knockout mice [83]. Additionally, tumor development was significantly delayed in p53 and HSF4 double-knockout mice when compared with p53-null mice, indicating an involvement of HSF4 in the proliferation of cancer cells. Considering that the knockout of either HSF1 or HSF4 alters the spectrum of tumors in p53-null mice, it is likely that both HSFs impact tumorigenesis in a tissue-specific manner. Although HSF1 knockdown impairs the proliferation of cancer cells, it has been reported to have only a minimal effect on the viability of non-cancer cells [79,84]. Intriguingly, HSF1 appears to be required for optimal proliferation of T cells in distinct tissues at non-febrile and febrile temperatures [85].

Numerous studies have shown that human cancer cells exhibit high expression of HSF1, which supports tumorigenesis [86]. Recently, a meta-analysis of patient samples revealed that HSF1 is overexpressed in different types of cancers mainly due to amplification of the chromosome region 8q24.3, wherein the gene encoding HSF1 resides [87]. In addition to HSF1, HSF2 and HSF4, whose coding region is located at 6q22.31 and 16q22.1 respectively, have also been indicated to impact malignant transformation [88,89,90]. Although there are only a few studies reporting on the expression of HSF2 and HSF4 in cancer, decreased expression of HSF2 has been shown in epithelial cancers, such as prostate and breast cancers as well as small-cell lung carcinomas, whereas low-grade gliomas, lung cancer tissue, and esophageal squamous cell carcinoma (ESCC) display high HSF2 expression [88,91,92,93]. To date, liver and colon cancer are the only types of cancer where elevated levels of HSF4 have been reported [90,94]. Hence, there is a great need for comprehensive analyses of HSF2 and HSF4 expression in different tumor types. Following chapters and Table 1 in this review will describe the currently available information of HSFs in specific cancer types.

### 5.1. HSF1 Cancer Signature

HSFs are known to be essential regulators of transcription in human cells exposed to acute stress, by inducing the expression of molecular chaperones, signaling molecules, and transcriptional regulators. Although HSFs share target genes under non-stress and stress conditions, they bind to distinct target loci in order to modulate gene expression [8,9,95]. During malignant transformation, an increased activity of HSF1 is necessary for maintaining high levels of HSPs to counteract the proteomic imbalance caused by genetic alterations, such as gene amplification [96,97]. However, the advanced genome-wide sequencing techniques have enabled identification of new gene regulatory networks that are driven by HSFs under severe pathological conditions, including cancer. Two studies stand out that demonstrated the importance of the HSF1-mediated transcriptional programs specific for malignant transformation and tumor progression [10,98]. The first study was published in 2012 by Mendillo and colleagues, who showed that HSF1 drives a cancer-specific transcriptional program in different types of tumors, which supports numerous oncogenic processes, such as cell-cycle regulation, metabolism, and adhesion. This transcriptional program, called the HSF1 cancer signature (HSF1-CaSig) differs from that induced by heat shock, since a major portion of the genes are uniquely regulated in cancer. Importantly, the HSF1-CaSig provides a predicting marker for cancer severity, since it was shown to correlate with poor patient survival in breast cancer patients [10]. Recently, a comparison of HSF1-CaSig in different cancer types revealed that numerous genes that are overexpressed, including HSF1, reside in a region of chromosome 8q, which is typically amplified in cancer [84,99].

The second study showed that cancer-associated fibroblasts (CAFs) require HSF1 for transcriptional reprogramming of the stroma [98]. CAFs are abundant in the tumor stroma, where they in a non-cell-autonomous manner enhance tumor progression by secreting cytokines and growth factors, which in turn support cell proliferation, angiogenesis, and invasion [100]. In CAFs, where HSF1 activity is high, HSF1 drives a transcriptional program that is distinct from the HSF1-CaSig in adjacent cancer cells [98]. This enables CAFs to mechanistically create a tumor-promoting microenvironment by increasing secretion of factors that have impact on gene expression, resulting in enhanced survival and proliferation of cancer cells [98]. Apart from HSF1, recent studies have identified HSF2 as an important regulator of gene networks that affects cancer progression and survival [88,101]. In human osteosarcoma U2OS cells, HSF2 is an essential survival factor in response to proteasome inhibition by maintaining the expression of cadherin superfamily genes and subsequent cell–cell adhesion [101]. During the progression of prostate cancer, a decrease in HSF2 expression has been observed, which could at least partly be due to cancer cells’ property to eliminate an HSF2-driven transcriptional program [88]. To date, however, no specific HSF2 cancer signature has been identified, thereby emphasizing that further studies on HSF2 in cancer are warranted.

### 5.2. HSFs in Breast Cancer

Breast carcinomas have been shown to overexpress HSF1 [102], and in this cancer type the HSF1-CaSig in patient samples is strongly associated with metastasis and death [10]. To date, several studies have characterized supportive oncogenic function for HSF1 in breast cancer, and multiple mechanisms contributing to HSF1 activation have been proposed (Figure 2). The transcription factor IER5, which is upregulated in the breast cancer cell lines MCF-7 and MDA-MB-231, was found to induce abnormal HSF1 activation, which subsequently promoted anchorage-independent cell growth [103]. Mechanistically, it was shown in 293T cells that IER5 forms a complex with protein phosphatase 2 (PP2A), which dephosphorylates HSF1, thus generating a hypo-phosphorylated form of HSF1 that induces upregulation of genes coding for HSPs. In estrogen receptor alpha (ERα) positive breast cancer cells, ERα stimulation promotes HSF1 activation through the MAPK/ERK pathway, in which the subsequent activation of the kinases MEK and ERK results in phosphorylation of HSF1 at serine 326 [104]. Phosphorylation of HSF1 at serine 326 is associated with HSF1 activation [105], and in ERα-positive breast cancer, HSF1 upregulates the expression of HSP90/*HSPC*, a chaperone that aids in the maturation of ERα and numerous kinases that drive tumorigenesis [82,106]. Hence, it is possible that a positive feedback loop engages ERα and HSF1 in ERα-positive breast cancers.

A key signaling nexus in breast cancer is the receptor tyrosine kinase HER2, also called ERBB2, which is upregulated in 25% of breast cancer cases and promotes metastasis, partially by activating HSF1 [82,107]. HSF1 activation in HER2-positive breast cancer depends on the ability of HER2 to constitutively stimulate phosphorylation of HSF1 at serine 326 by protein kinase B (PKB), also called AKT [108,109]. AKT is one of the major kinases in the phosphoinositide 3-kinase (PI3K)/AKT pathway that modulates several biological processes including growth and survival [110]. Once activated, HSF1 upregulates the expression of HSPs, e.g., HSP90/*HSPC*. Additionally, HSF1 facilitates epithelial-mesenchymal transition (EMT) by upregulating the expression of the transcription factor SLUG [109]. The HER2-AKT-HSF1 axis maintains high HSP90/*HSPC* expression, which is critical for sustaining the stability and activity of mutated or overexpressed kinases and transcription factors. This has been demonstrated for mutant p53, HER2, and AKT as well as for RAF1, which is a kinase in the MAPK/ERK pathway that can promote cancer cell survival and tumor development [82,111,112]. Surprisingly, a recent report showed that HSF1 activity is high in HER2-positive breast cancer cells that are resistant to Lapatinib, a drug used to inhibit HER2. Therefore, it is likely that during breast cancer development other signaling pathways activate HSF1 in order to compensate for the loss of HER2 function [113].

Recently, HSF1 was shown to promote mammary tumorigenesis by enhancing DNA repair [114]. In a breast cancer model, comprising several cancer cell lines, HSF1 forms a ternary complex with poly-(ADP-ribose) polymerases 1 and 13 (PARP1 and PARP13), which are key facilitators of DNA repair pathways, and in response to DNA damage HSF1 and PARP13 aid in the activation of PARP1 by supporting its auto-PARylation. Once activated, PARP1 is redistributed to the sites of double-stranded breaks, which in turn improves DNA repair efficiency likely through the recruitment of DNA repair factors [114]. Apart from DNA repair, HSF1 indirectly modulates the expression of β-catenin, a pivotal component of the Wnt signaling pathway that affects numerous aspects in tumorigenesis, such as immunity and cancer stem cell maintenance [115,116]. HSF1 enhances β-catenin levels by reducing the expression of lincRNA-p21 and increasing that of the RNA-binding protein HuR, which are known to be involved in the translational regulation of β-catenin mRNA [115].

The functional impact of HSF2 on breast cancer has only recently started to emerge (Figure 2). In breast cancer cells, HSF2 and the transcription factor zinc finger E-box-binding homeobox 1 (ZEB1) enhance tumorigenesis by cooperatively stimulating the expression of the microRNA cluster miR-183/-96/-182 [117]. miR-183/-96/-182 is transcribed in one pri-miRNA, and once processed, each individual miRNA promotes cell proliferation and migration in the breast cancer cell lines MCF-7 and T47D [117]. miR-183 has been reported to downregulate the small GTPase RAB21, and consequently support aneuploidy, which is known to increase genetic heterogeneity and tumor evolution [118]. HSF2 has also been shown to modulate the expression of the enzyme Dol-P-Man:Man(5)GlcNAc(2)-PP-Dol alpha-1,3-mannosyltransferase (ALG3) in breast cancer [119]. High ALG3 expression, as maintained by HSF2, increased both proliferation and migration of MCF-7 cells. Furthermore, in nude mice injected with MCF-7 cells, the removal of ALG3 from the cancer cells reduced both tumor development and HSF2 levels, which indicates a positive feedback between ALG3 and HSF2 [119].

It remains to be elucidated whether HSF4 directly impacts breast cancer development. Nevertheless, both HSF4 and HSF2 have been indicated to modulate the expression of hypoxia-inducible factor 1-alpha (HIF-1α) in MCF-7 breast cancer cells [120] (Figure 2). HIF-1α is the master regulator of transcription in response to hypoxic stress, and in malignant tissues HIF-1α has been shown to increase the levels of vascular endothelial growth factor (VEGF), a factor known to stimulate blood vessel formation [121]. Knockout or overexpression of either HSF2 or HSF4 in MCF-7 cells increased the levels of HIF-1α and VEGF, indicating that these HSFs are required to maintain a steady expression of HIF-1α. In accordance, HSF1 depletion in MCF-7 cells was shown to reduce the levels of HIF-1α and its target VEGF, which was at least partly dependent on HSF1’s ability to stimulate the expression of the mRNA-binding protein HuR that controls mRNA stability and translation of numerous proteins involved in cancer, including HIF-1α and VEGF [122]. Taken together, it is possible that increased expression of a single HSF, or multiple members of the HSF family enhances fundamental breast cancer-promoting processes, such as angiogenesis.

### 5.3. HSFs in Hepatocellular Carcinoma (HCC)

Similarly to many other cancers types, HCC patient samples and cell lines, including SM7721 and M7024, display high levels of HSF1 protein and enhanced phosphorylation at serine 326 [123]. The mammalian target of rapamycin (mTOR) has been shown to phosphorylate HSF1 at serine 326, which contributes to the activation of HSF1 in HCC [124], but the upstream signaling pathways are poorly delineated. Recently, HSF1 depletion in HLE and HLF hepatoma cells was reported to result in enhanced apoptosis, reduced proliferation, lipid depletion, and decreased glycolysis [125]. At the molecular level, lack of HSF1 decreased many key proteins in the PI3K/AKT/mTOR cascade, a signaling pathway known to modulate cell growth and survival under physiological and pathological conditions. The potential clinical relevance of HSF1 in HCC is supported by the findings that the abolishment of HSF1 in mice overexpressing AKT, inhibited AKT-driven hepatocarcinogenesis [125]. Nevertheless, the precise mechanism by which HSF1 sustains the oncogenic potential of the PI3K/AKT/mTOR cascade remains to be determined. Interestingly, HSF1 has been shown to promote migration and invasion of HCC cells by directly enhancing the expression of miR-135b, which is characteristically amplified and upregulated in HCC tissues [126,127] (Figure 3). In SMMC-7721 and Huh-7 HCC cell lines, overexpression of miR-135b markedly reduced the expression of the target proteins RECK and EVI5, of which only RECK had previously been shown to suppress migration of cancer cells [127].

In addition to HSF1, HSF2 is also overexpressed in Huh-7 and SMMC-7721 HCC cells, where it mediates extensive metabolic reprogramming that allows cancer cells to modulate their energy production requirements [89,128]. In Huh-7 and SMMC-7721 cells, HSF2 was shown to interact with euchromatic histone lysine methyl transferase 2 (EHMT2) in order to epigenetically silence the gene encoding fructose-bisphosphatase 1 (FBP1), which is a tumor suppressor and negative regulator of aerobic glycolysis [89] (Figure 3). This silencing of FBP1, mediated by HSF2 and EHMT2, promotes aerobic glycolysis and thereby stimulates cell proliferation. HSF2 was also shown to be required for maintaining the expression of glucose transporter 1 (GLUT1), hexokinase 2 (HK2), and lactate dehydrogenase A (LDHA), all of which are important regulators of glycolysis [89]. Although it is not yet known how HSF2 regulates these genes, there is some evidence for HSF2-driven activation of HIF-1α, which is the main transcriptional regulator of GLUT1, HK2, and LDHA in hypoxia [129]. Furthermore, overexpression of FBP1 can suppress the expression of HIF-1α in MDA-MB-468 breast cancer cells exposed to hypoxic conditions [130]. Therefore, it is tempting to speculate that HSF2 stimulates aerobic glycolysis in HCC through two separate mechanisms, i.e., by epigenetically silencing FBP1 and by inducing HIF-1α expression (Figure 3).

HSF4 was recently found to be overexpressed in primary HCC tissues and high HSF4 expression correlated with poor survival of HCC patients [90]. In Huh-7 and SMMC-7721 cells, HSF4 was shown to stimulate the proliferation, migration, and invasion capacities by triggering EMT, which required HIF-1α-dependent activation of AKT [90]. Since previous results had already indicated that HSF4 can modulate the expression of HIF-1α in MCF-7 breast cancer cells [120], it is possible that, similarly to HSF2, the function of HSF4 is hijacked in HCC in order to enhance HIF-1α expression, thereby allowing cancer progression.

### 5.4. HSF1 and HSF2 in Prostate Cancer

One of the first studies describing the function of HSF1 in cancer showed that HSF1 contributes to the development of polyploidy in prostate carcinoma [131]. Overexpression of HSF1 enhanced the development of polyploidy, whereas reduction in HSF1 activity partially restored diploid DNA content in PC-3 cells, a human prostate cancer cell line that neither expresses the androgen receptor nor the prostate-specific antigen [132]. Polyploidy is a common feature in cancer cells and it can enhance tumor progression [133]. Recently, a comprehensive study by Björk and colleagues proposed that HSF1 is a strong predictive biomarker in prostate cancer [134]. In that study, a clinical prostate cancer dataset displayed high HSF1 mRNA expression, and the immunohistochemically analysis of tissue micro-arrays showed a dramatic increase in nuclear HSF1 in samples derived from patients suffering from advanced prostate cancer. Furthermore, patients presenting elevated levels of nuclear HSF1 and high Gleason score (a system used to characterize prostate biopsies) were likely to require secondary therapy and generally displayed poor disease-specific survival which highlights a positive correlation between dose-dependent expression of HSF1 and Gleason score.

In contrast to HSF1, HSF2 appears to function as a tumor suppressor in prostate cancer [88]. The expression of HSF2 is low in most of the prostate cancer tissues, which is mainly due to a heterozygous loss of HSF2, and low HSF2 expression correlates with increased metastasis and poor patient survival as well as a high Gleason score. Moreover, in three-dimensional organotypic cultures and in the in vivo xenograft chorioallantoic membrane model, knockdown of HSF2 from PC-3 cells enhanced organoid differentiation and promoted invasive growth. While the molecular mechanism by which HSF2 suppresses prostate cancer development remains to be established, a gene expression profiling indicated that HSF2 modulates GTPase activity, cell–cell adhesion, and actin cytoskeleton dynamics, all of which have altered expression in cancer [88]. Importantly, a meta-analysis demonstrated that in addition to prostate cancer, the mRNA expression of HSF2 is downregulated in a variety of cancer tissues [88], suggesting that HSF2 downregulation is a key event during cancer development.

### 5.5. HSF1 and HSF2 in Lung Cancer and Esophageal Squamous Cell Carcinoma (ESCC)

HSF1 expression has been reported to be upregulated in non-small cell lung cancer and it is associated with increased angiogenesis and poor patient survival [135]. In lung cancer, CAFs can also promote the activity of HSF1, which stimulates tumor progression and metastasis [98]. Similarly to HSF1, the expression of HSF2 has been found to be upregulated in tumor tissues from lung cancer patients [92]. Although it is not yet known by which mechanisms HSF2 expression is enhanced in lung cancer cells, there is a disease-specific correlation between the high expression of HSF2 and the transcription factor USF2 in A549 lung cancer cells [92,136]. Reporter assays, in which several fragments of the HSF2 promoter were characterized, have shown that the E-Box element, to which USF2 binds, is essential for driving the transcription of HSF2 in rat neuronal C6 cells [137]. Hence, it is possible that USF2 stimulates HSF2 expression also in lung cancer. In non-tumorigenic BEAS-2B cells and A549 lung cancer cells, the elevated levels of HSF2 promote cell proliferation and migration, which is partially dependent on HSF2’s ability to induce the expression of HSP27/*HSPB1* and HSP90/*HSPC* [92].

In ESCC, both HSF1 and HSF2 are upregulated, which correlates with increased expression of HSPs [93,138]. Based on the analyses of ESCC patient samples, abundantly expressed HSF1 is associated with poor patient survival and increased expression of HSP27/*HSPB1* and HSP90/*HSPC* [138]. In ESCC, HSF2 mainly induces the expression of HSP70/*HSPA1A*, which in turn suppresses apoptosis by preventing caspase 3 activation. Collectively, these results demonstrate the importance of elevated levels of HSPs in the development of ESCC. Interestingly, high expression of HSF2 in ESCC is due to the downregulation of miR-202, a microRNA that targets the 3′-UTR in HSF2 mRNA [93]. In contrast to these results, two studies have suggested that elevated HSP70/*HSPA1A* expression in ESCC generally correlates with improved patient survival [139,140]. Therefore, it is possible that HSF2 can counteract tumor progression in certain cases of ESCC.

### 5.6. HSF1 and HSF4 in Colorectal Cancer (CRC)

Colorectal cancer tissue and colorectal cancer cell lines, including HCT116, SW480, and SW620, display elevated levels of HSF1, and one of its main tumor-promoting functions is to enhance glutaminolysis [141]. Glutaminolysis is a metabolic pathway required for macromolecular synthesis, and glutamine deprivation has been shown to reduce the growth of colorectal cancer cells [142]. Mechanistically, HSF1 recruits the DNA methyltransferase DNMT3a to the host gene encoding mir137, which targets glutaminase 1 (GLS1) mRNA [141]. GLS1 is the rate-limiting enzyme for converting glutamine to glutamate, and its activation is partially dependent on mTOR [141]. DNMT3a can epigenetically silence mir137, resulting in elevated levels of GLS1. Consequently, silencing of HSF1 in HCT116, SW480, and SW620 cells decreased both mTOR activity and glutaminolysis, which subsequently reduced cell growth [141]. Apart from metabolic reprogramming, HSF1 has been shown to support the growth of CRC cells exposed to stress by upregulating co-chaperones, such as BAG3, which protects CRC cells by stabilizing antiapoptotic Bcl-XL, Mcl-1, and Bcl-2 [143].

Recently, HSF4 was found to be abundantly expressed in patients with primary CRC [94]. This original finding described the clinical relevance of HSF4 in cancer and showed that high expression of HSF4 correlates with advanced disease progression, cancer recurrence and poor patient survival. Based on the bioinformatics analyses, it was suggested that HSF4 directly interacts with the protein phosphatase DUSP26, which has previously been shown to modulate the phosphorylation status of HSF4 and decrease its activity [144]. The direct function of HSF4 in CRC remains to be investigated. Considering that HSP27/*HSPB1* and HSP90/*HSPC* have many pro-tumorigenic properties in different cancer types [78], it would be important to determine whether HSF4 plays a key role in regulating the expression of HSP27/*HSPB1* and HSP90/*HSPC* in CRC.

## 6. Conclusions

Studies revealing that HSFs regulate distinct transcriptional programs under specific circumstances, both physiological and pathological, underscore an emerging theme that the target genes of HSFs play critical roles in many developmental and differentiation processes, cell–cell adhesion and motility as well as in tumorigenesis. In particular, HSF1, the most studied member of the HSF family, provides a node in complex regulatory networks that drives CaSig in different malignant tissues, which promotes progression of tumorigenesis. Given the recent reports on other members of the HSF family influencing the growth and other properties of cancer cells, it will be important to conclusively map the genomic loci that are regulated by HSF2 and HSF4, either alone or in conjunction with HSF1 and other transcriptional regulators. This type of multi-faceted approach is critical for the elucidation of how transcription factors cooperate in order to drive cancer progression. Furthermore, it is important to emphasize that several properties of HSFs in cancer, including their expression levels and cellular localization, support the notion that these transcription factors could be used as biomarkers for specific cancer types, especially in the contexts of personalized medicine.

Several drugs targeting HSF1 have already been and are being developed, which is an exciting topic recently summarized in a comprehensive review by Dong and colleagues [86]. Currently, however, no HSF1 inhibitor is cleared for clinical use due to fundamental gaps in knowledge of drug-molecular engagement with HSF1, pharmacokinetics and off-target effects. Forthcoming studies will show their efficacy and specificity, and whether any of these drugs will be approved by the FDA. Considering that HSF1, HSF2, and HSF4 display multiple properties in cancer, which are either unique or common for each factor, or whether they act synergistically or antagonistically with each other, the need for new and improved agents targeting HSFs is significant. It can be hypothesized that future HSF inhibitors may display synergistic therapeutic effects in combination with drugs such as the proteasome inhibitor Bortezomib, which is used in the treatment of multiple myeloma. The disruption of both HSP synthesis and general protein degradation, will likely have an additive effect on eliminating cancer cells, and might even aid in the prevention of drug resistance, which commonly develops in multiple myeloma patients treated with Bortezomib [145].

Further endeavors in drug development should focus on the attenuation phase of the heat shock response, which is paradoxically hampered in cancer cells despite the elevated levels of HSPs. Adding to the complexity of the regulatory mechanisms, future studies should explore to which extent specific HSPs are expressed in the nucleus and how they affect the activity of HSFs. A key approach in targeting the heat shock response in cancer might be through enhanced attenuation. Intriguingly, this approach is supported by results indicating that histone de-acetylase inhibitors (HDACi) can stimulate the attenuation phase of the heat shock response [48]. The anti-cancer effects of HDACis [146], raise a possibility that they could, most probably indirectly, increase acetylation of HSF1, which in turn has been shown to stimulate de-activation of HSF1 and thereby attenuate the heat shock response [48]. Fundamental questions regarding the underlying mechanisms of the activation-attenuation cycle of HSFs need to be addressed prior to the development of novel therapeutic approaches targeting cancer and other pathological conditions.

## Figures and Tables

**Figure 1 cells-09-01202-f001:**
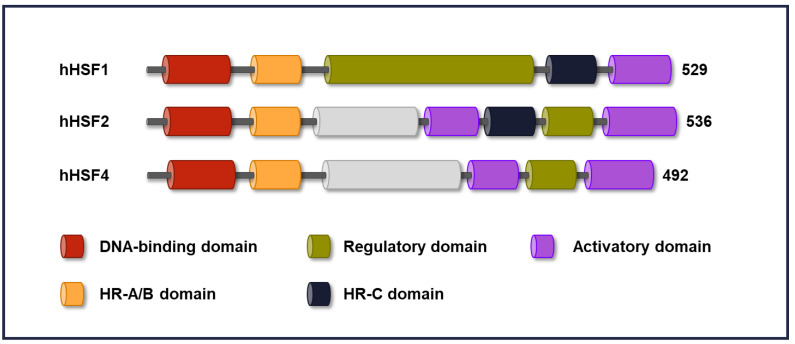
Domain organization of human heat shock transcription factors (HSFs). Schematic illustration of human HSF1 (hHSF1), HSF2 (hHSF2), and HSF4 (hHSF4) with the known functional domains. The number indicates the last amino acid of each protein. Each HSF contains a winged helix–turn–helix DNA-binding domain and an oligomerization domain (HR-A/B). The HR-C domain in HSF1 and HSF2 suppresses HSF oligomerization and keeps them inactive under normal conditions. HSFs modulate transcription via the activatory domain, and in HSF1 the regulatory domain controls stress responsiveness. HSF1 and HSF2 display ~39% identical amino acid sequence, whereas HSF1 and HSF4 display ~42% similarity. Note that the figure is not drawn to scale.

**Figure 2 cells-09-01202-f002:**
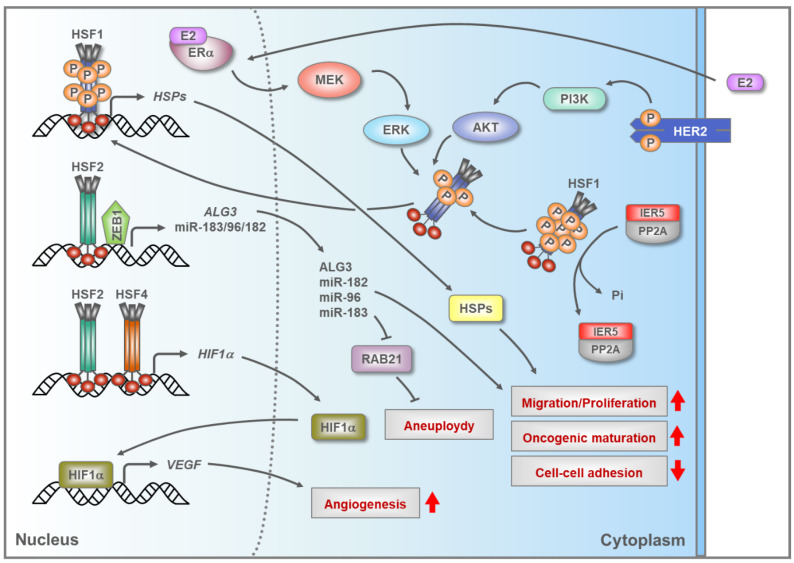
Breast cancer cells use HSFs to drive various tumor-promoting mechanisms. Several mechanisms contribute to the activation of human HSFs in breast cancer. Upon activation, trimeric HSF1, HSF2 and HSF4 bind to DNA and initiate the transcription of their target genes, which are involved in tumorigenesis. Multiple proteins enhance the activation of HSF1 by regulating its phosphorylation status. The hormone 17β-estradiol (E2) binds to and activates estrogen receptor alpha (ERα), which leads to HSF1 phosphorylation through the mitogen-activated protein kinases/extracellular signal-regulated kinases (MAPK/ERK) pathway, in which the kinases MEK and ERK increase the phosphorylation of HSF1, most probably in the cytoplasm. Activation of HSF1 is also enhanced by epidermal growth factor receptor 2 (HER2), which increases the phosphorylation of HSF1 by the intermediate kinases phosphatidylinositol 3-kinase (PI3K) and protein kinase B (AKT). Phosphorylated residues in HSF1 that are known to repress the trans-activating capacity of HSF1, are dephosphorylated (Pi) by a complex consisting of the transcription factor immeadiate early response gene 5 protein (IER5) and protein phosphatase 2 (PP2A), thereby promoting HSF1 activation. In the nucleus, once activated, HSF1 induces the transcription of genes encoding heat shock proteins (HSPs), which enhance migration, proliferation, and oncogenic maturation as well as decrease cell–cell adhesion. HSF2 and the transcription factor zinc finger E-box-binding homeobox 1 (ZEB1) cooperatively induce the transcription of pri-mir-183/96/182, and following post-transcriptional processing, these miRNA molecules stimulate migration and proliferation. For example, mir-183, targets the small GTPase RAB21, a protein that prevents aneuploidy. HSF2 also stimulates the expression of Dol-P-Man:Man(5)GlcNAc(2)-PP-Dol alpha-1,3-mannosyltransferase (ALG3), an enzyme involved in modulating migration and proliferation in breast cancer cells. HSF2 together with HSF4 modulates the transcription of hypoxia-inducible factor 1 alpha (HIF-1α), a transcription factor regulating a myriad of genes, including vascular endothelial growth factor (VEGF) that promotes angiogenesis. Abbreviation: P, phosphorylation.

**Figure 3 cells-09-01202-f003:**
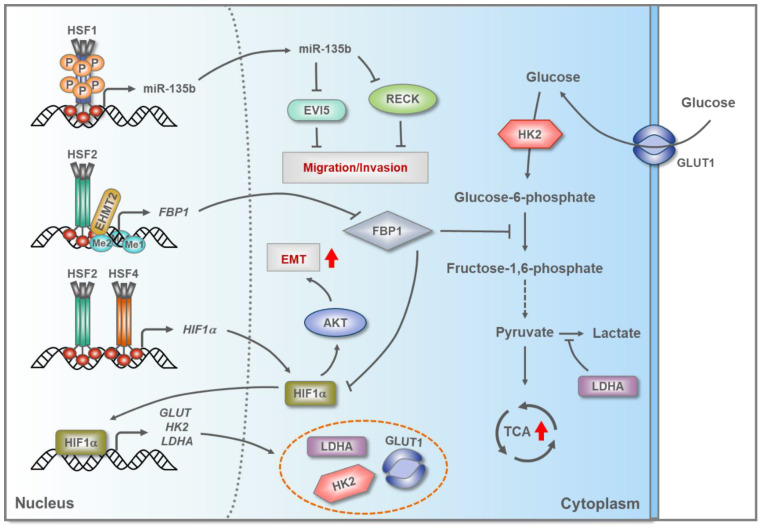
Hepatocellular carcinoma (HCC) cells use HSFs to promote metabolic reprograming and cancer invasion. In HCC, trimeric human HSF1, HSF2 and HSF4 bind to DNA and induce the transcription of genes that modulate multiple biological processes that support tumorigenesis. HSF1 enhances the expression of miR-135b that targets reversion-inducing cysteine-rich protein with Kazal motifs (RECK) and ecotropic viral integration site 5 protein homolog (EVI5), both of which are repressors of migration and invasion. Together, HSF2 and euchromatic histone lysine methyl transferase 2 (EHMT2) promote methylation of the gene encoding fructose-bisphosphatase 1 (FBP1), resulting in a decreased expression of FBP1, an enzyme that can counteract glycolysis and inhibit the expression of hypoxia-inducible factor 1 alpha (HIF-1α). Together with HSF4, HSF2 modulates the expression of HIF-1α, which regulates a myriad of genes, including glucose transporter GLUT1, hexokinase-2 (HK2), and l-lactate dehydrogenase A (LDHA), all of which are involved in various steps of the glycolysis pathway in HCC. HIF-1α can also facilitate the activation of protein kinase B (AKT), which promotes epithelial-mesenchymal transition (EMT). Abbreviations: Me1/Me2, mono/di-methylation; P, phosphorylation; TCA, tricarboxylic acid cycle.

**Table 1 cells-09-01202-t001:** Summary of HSFs expression and function in different types of cancer.

HSF	Cancer Type	Expression in Cells and Tissues	Effect on Tumorigenesis
HSF1	HCC	High	Promotes cell proliferation, growth, migration, invasion, and survival as well as kinase function, lipid metabolism, and glycolysis.
HSF2	HCC	High	Promotes cell proliferation and aerobic glycolysis.
HSF4	HCC	High	Promotes cell proliferation, migration, kinase function, and EMT.
HSF1	Breast cancer	High	Promotes cell motility, metastasis, and survival as well as receptor and kinase maturation, stemness, drug resistance, DNA repair, and EMT.
HSF2	Breast cancer	High	Promotes cell proliferation, migration, and aneuploidy.
HSF1	Prostate cancer	High	Promotes development of polyploidy, high Gleason score, and cancer re-occurrence.
HSF2	Prostate cancer	Low	Promotes organoid differentiation, invasive growth, and high Gleason score.
HSF1	Lung Cancer	High	Decreases patient survival. Promotes angiogenesis and metastasis.
HSF2	Lung Cancer	High	Promotes cell proliferation, migration, and expression of HSPs.
HSF1	ESCC	High	Promotes cell survival and expression of HSPs.
HSF2	ESCC	High	Promotes cell survival and expression of HSPs.
HSF1	CRC	High	Promotes expression of anti-apoptotic proteins, cell growth, and glutaminolysis.
HSF4	CRC	High	Decreases patient survival. Promotes cancer re-ocurrence.

Abbreviations: HCC, hepatocellular carcinoma; ESCC, esophageal squamous cell carcinoma; CRC, colorectal cancer.

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
