# Peer review of "Molecular Mechanisms of Heat Shock Factors in Cancer"

_cells, 2020, doi:10.3390/cells9051202_

Round 1

Reviewer 1 Report

In this review article, Mikael Christer Puustinen and colleagues provided a nice review of the molecular mechanisms of heat shock factors. This review is well written with sufficient reference to support this article. This review discussed the expression patterns and functions of HSF1, HSF2, and HSF4 following the role of heat shock factors in normal conditions. Especially, the authors provide a clear review of the heat shock factors related pathways in breast cancer and liver cancer. There is no additional revision needed. Congratulation for this informative review.

Author Response

Reviewer 1

Thank you for your positive and constructive comments. No corrections were needed.

Reviewer 2 Report

The review submitted by Puustinen and Sistonen is  well written summary of HSF and HSP in biology and in cancer. Some comments are as follows:

Line 49-50: Perhaps the authors can comment on HSF sequence identity within human genes and compared to more ancient species (e.g. insects) as to whether gene duplication followed by divergence occurred. Also, are the human HSF genes on different chromosomes or are they in a cluster on one chromosome? Any located on sex chromosomes in mice or humans (should probably give chromosome numbers for mice and humans since this is closely followed in genetics).

Line 62: “HSF1 contains also a regulatory” should be “HSF1 also contains a regulatory”

Line 80: “viruses” , do other intracellular pathogens (intracellular bacteria, parasites) also induce HSFs? If so, perhaps “pathogens” is more encompassing.

Line 82: “Although the heat shock response appears to be a universal defense response in all organisms the primary mechanism initiating the response still remains unknown”. Not all organisms, but most -> see PMID10887071

Line 98-99: C. elegans and S. cerevisiae and others…at first mention the full genus species name should be given in italics. Afterwards the italicized C. elegans or S. cerevisiae term can be used.

Line 108: how do the cytosolic HSF1 molecules translocate to nucleus? Bind to nuclear localization containing chaperones? Is dimerization or trimerization required for nuclear translocation, or can monomers transclocate?

Line 123: “been suggested to modulate also the attenuation phase” should read “been suggested to also modulate the attenuation phase”

Lines 136-144: is anything known regarding how infections may alter the proteosomal degradation of HSF1 and 2? Since interferons (IFNgamma) generagted during infections (particularly viral) alter proteosomal selectivity, this effect may skew the selection away from HSF1 and 2 that are presumably needed to fight infection.

Line 149: italicize Drosophila melanogaster

Line 158: are any double or triple KO mice reported? If not, does the author think that some combinations may be embryonic lethal?

Line 218: HSF1 KO does not significantly affect proliferation of normal cells, but does impact cancer cells. What about T cell proliferation? These ‘normal’ cells share many features with cancer cells like proliferative burst, redox properties, energy demands (aerobic glycolosis, Warburg effect), migration/metastasis, etc. I would imagine HSG1 KO T cells do not proliferate well when stimulated through the T cell receptor. Has this been studied? If not, the authors need to mention this.

Line 231: What about expression of HSFs during increased stages of one type of cancer? Is anything known? If not, author should mention this gap in knowledge.

Author Response

Reviewer 2

Thank you for your critical and constructive comments and suggestions. Below, please find the point-by-point reply to all the concerns.

Line 49-50

Perhaps the authors can comment on HSF sequence identity within human genes and compared to more ancient species (e.g. insects) as to whether gene duplication followed by divergence occurred. Also, are the human HSF genes on different chromosomes or are they in a cluster on one chromosome? Any located on sex chromosomes in mice or humans (should probably give chromosome numbers for mice and humans since this is closely followed in genetics).

Despite the wealth of research on HSF biology, the evolutionary origins of the genes encoding HSFs are not conclusively explored. Invertebrates, such as insects express only one HSF, and for example Drosophila melanogaster (fruit fly) HSF displays ~49% identical amino acid sequence to that of human HSF1. Based on InsectBase, human HSF1 displays ~58% identical sequence to that of HSF in Zootermopsis nevadensis (termite). Surprisingly, Drosophila melanogaster HSF is only ~56% identical to that of Zootermopsis nevadensis, and in both cases it appears that two distinct domains are highly similar to the corresponding regions in human HSF1. The sequence comparisons among human HSFs reveal that both HSF2 and HSF4 differ significantly from HSF1 (now indicated in the figure legend of Figure 1, lines 73-74). It is also interesting to note that lizards display gene sequences that are similar to those of human HSF1, HSF2 and HSF4. Taken together the evidence that is available in literature and databases, it is possible that the genes encoding HSF2 and HSF4 originate from a gene duplication of HSF1, but it is unclear at which stage during evolution this occurred. Within the scope of this review, we have decided not to discuss the evolutionary aspects of HSF genetics in a great detail, as they have been previously reviewed, see Ref. 2 and references therein.

Each HSF is located in different murine and human chromosomes (lines 162-165 and lines 228-229).

Line 62:

“HSF1 contains also a regulatory” should be “HSF1 also contains a regulatory”

Thank you for indicating this section of text that needed improvement. This is now corrected (line 62).

Line 80:

“viruses” , do other intracellular pathogens (intracellular bacteria, parasites) also induce HSFs? If so, perhaps “pathogens” is more encompassing.

Thank you for indicating this section of text that needed improvement. It is true that other organisms including bacteria and parasites also induce the HSR, and we have replaced the word “viruses” with “pathogens” (line 80).

Line 82:

“Although the heat shock response appears to be a universal defense response in all organisms the primary mechanism initiating the response still remains unknown”. Not all organisms, but most -> see PMID10887071

Thank you for pointing out this issue. The mistake is corrected (lines 82-83).

Line 98-99:

  1. elegans and S. cerevisiae and others…at first mention the full genus species name should be given in italics. Afterwards the italicized C. elegans or S. cerevisiae term can be used.

Thank you for bringing this mistake to our attention, which we have now corrected (lines 100-101, 105).

how do the cytosolic HSF1 molecules translocate to nucleus? Bind to nuclear localization containing chaperones? Is dimerization or trimerization required for nuclear translocation, or can monomers transclocate?

These questions are very relevant for understanding the fundamental regulatory mechanisms of the HSR. To the best of our knowledge, it is still unknown if dimers or trimers can pass the nuclear envelope or whether monomers predominantly translocate into the nucleus prior to trimerization. It has been reported that under non-stress and stress conditions, HSF1 nuclear accumulation is promoted by a potent import signal (Vujanac et al. 2005). The same study showed that stress inhibits the export competence of HSF1, thereby resulting in a significant nuclear enrichment, which is now mentioned in the text (lines 84-85). These results indicate that the nuclear export is the main determining factor regulating HSF1 levels in the nucleus upon exposure to stress. The exact mechanisms by which HSPs modulate HSF activity in the nucleus are unknown, and it remains to be established if specific HSPs are required for the functions of HSFs in the nucleus. This issue is presented in the text (lines 529-530).

Vujanac M, Fenaroli A, Zimarino V. Constitutive nuclear import and stress-regulated nucleocytoplasmic shuttling of mammalian heat-shock factor 1. Traffic. 2005, 6, 214-229.

Line 123:

“been suggested to modulate also the attenuation phase” should read “been suggested to also modulate the attenuation phase”

Thank you for indicating this section of text needed improvement, which is corrected (line 125).

Lines 136-144:

is anything known regarding how infections may alter the proteosomal degradation of HSF1 and 2? Since interferons (IFNgamma) generagted during infections (particularly viral) alter proteosomal selectivity, this effect may skew the selection away from HSF1 and 2 that are presumably needed to fight infection.

To the best of our knowledge, it is unknown if infections directly affect the degradation of HSF1 and HSF2. However, it is interesting to speculate that the cellular stress caused by infections might induce the proteasomal degradation of HSF2 in a similar manner as described for heat stress (lines 143-146). It has been indicated that cytokines (e,g, IL-6 and IFN-γ), released upon infection, stimulate the induction of HSPs via activating the STAT transcription factors. Importantly, HSF1 interacts with STAT1 or STAT3 to modulate the transcription of HSPs, suggesting that HSFs interact with other transcription factors in order to cooperatively regulate gene expression (Stephanou & Latchman 1999). For this reason, it is likely that the expression of HSF1 is not decreased during infections.

Stephanou A, Latchman DS. Transcriptional regulation of the heat shock protein genes by STAT family transcription factors. Gene Expr. 1999, 7, 311-9.

Line 149:

italicize Drosophila melanogaster

This mistake is corrected (line 151).

Line 158:

are any double or triple KO mice reported? If not, does the author think that some combinations may be embryonic lethal?

To the best of our knowledge, no triple knockout HSF mice have been generated. Although HSF double-knock mice are viable, the double knock-out of HSF1 and HSF2 resulted in male sterility (Wang et al., 2004; Jin et al., 2018) which is discussed in the text (line 177). Considering that the expression of HSF4 is mainly limited to the eye under normal conditions and that HSF1-HSF2 double-null mice are viable, it is possible that mice lacking all three HSFs would also be viable, but we do not want to speculate too much due to a lack of evidence in literature.

Wang, G.; Ying, Z.; Jin, X.; Tu, N.; Zhang, Y.; Phillips, M.; Moskophidis, D.; Mivechi, N.F. Essential requirement for both hsf1 and hsf2 transcriptional activity in spermatogenesis and male fertility. Genesis 2004, 38, 66-80.

Jin,X.; Eroglu, B.; Moskophidis, D.; Mivechi, N.F. Targeted deletion of Hsf1, 2, and 4 genes in mice. Methods Mol. Biol. 2018, 1709, 1-22.

Line 218:

HSF1 KO does not significantly affect proliferation of normal cells, but does impact cancer cells. What about T cell proliferation? These ‘normal’ cells share many features with cancer cells like proliferative burst, redox properties, energy demands (aerobic glycolosis, Warburg effect), migration/metastasis, etc. I would imagine HSG1 KO T cells do not proliferate well when stimulated through the T cell receptor. Has this been studied? If not, the authors need to mention this.

The role of HSFs in modulating the immune system is of great importance. Early studies indicated that HSF1 has a key role in these processes as HSF1-null mice have impaired IgG production. Subsequently, several reports have shown that HSF1 is involved in regulating T cell proliferation and function. In the absence of HSF1, T cell proliferation is hypersensitive to proteotoxic stress, which is especially relevant at febrile temperatures. Furthermore, already at sub-febrile temperatures, upon T cell receptor ligation and activation, the lack of HSF1 impacts T cells proliferation. For example, HSF1-null T cells show an impaired proliferation in response to staphylococcal enterotoxin B (SEB) in spleen. However, it is important to note that the T cell model systems and treatments can affect the sensitivity of assays performed under non-stress conditions. Future studies are warranted to increase our understanding of the impact of HSF1 in T cell biology.

The effects of HSF1 on T cell proliferation under non-stress and stress conditions are now mentioned in the text (lines 221-224).

Gandhapudi SK, Murapa P, Threlkeld ZD, Ward M, Sarge KD, Snow C, Woodward JG. Heat shock transcription factor 1 is activated as a consequence of lymphocyte activation and regulates a major proteostasis network in T cells critical for cell division during stress. J Immunol. 2013,191, 4068-4079.

Line 231:

What about expression of HSFs during increased stages of one type of cancer? Is anything known? If not, author should mention this gap in knowledge.

Thank you for reminding us to emphasize this issue. One study demonstrating the dose-dependent effect of elevated HSF1 expression in cancer is now highlighted (lines 432-433). Since similar studies characterizing the expression levels of HSF2 and HSF4 in tumor samples are still scarce, it is not known to which degree these HSFs are expressed during the different phases of cancer progression.

Reviewer 3 Report

I reviewed the manuscript by Puustinen and Sistonen entitled: „Molecular Mechanisms of Heat Shock Factors in Cancer”. Heat Shock Factors regulate transcriptional programs that enable cellular adaptation to proteotoxic stress conditions. Moreover, they play an important role in other physiological processes associated with development and differentiation. In past decades its role in tumorigenesis has also been proved.

In my opinion, the manuscript is well written. In the beginning, the authors describe the structure and mechanisms regulating activity  of  HSFs. Next sections are concentrated on the role of HSFs in cancer. This manuscrpt summarizes available data reffering the role of HSF1, HSF2 and HSF4 in specific cancer types. The role of HSF1 in tumorigenesis is the most studied, the role of HSF2 and HSF4 is less understood. So far, there are numerous reviews referring the role of HSF1 in cancer. The contribution of HSF2 and HSF4 to tumorigenesis is a novelity of this review.  

I propose that authors could highlight  the different pattern of expression HSF1, HSF2 and HSF4 in different type of cancer and its potential role by summarizing the data in table, when it is possible. For example, high nuclear HSF1 was detected in advanced prostatÄ™ cancer and correlated with poor disease-specific survival. While HSF2 appears to function as a tumor suppressor in this type of cancer.

Author Response

Reviewer 3

Thank you for the critical and constructive comments and suggestions.

As requested, we have added a table to summarize the expression and function of human HSFs in different cancer types (Table 1).